palaeontology

conodont white matter, cSAXS, EBSD, X-ray tomography, ptychographic nanotomography

**Authors for correspondence:**
Ayse Atakul-Özdemir
e-mail: aozdemir@yyu.edu.tr
Carlos Martínez-Pérez
e-mail: carlos.martinez-perez@uv.es
Philip C. J. Donoghue
e-mail: phil.donoghue@bristol.ac.uk

# X-ray nanotomography and electron backscatter diffraction demonstrate the crystalline, heterogeneous and impermeable nature of conodont white matter

Ayse Atakul-Özdemir[1,2], Xander Warren[3],
Peter G. Martin[3], Manuel Guizar-Sicairos[4], Mirko Holler[4],
Federica Marone[4], Carlos Martínez-Pérez[2,5] and
Philip C. J. Donoghue[2]

[1]Department of Geophysical Engineering, Yuzuncu Yil University, 65180 Van, Turkey
[2]School of Earth Sciences, University of Bristol, Life Sciences Building, Tyndall Avenue Bristol, Bristol BS8 1TQ, UK
[3]Interface Analysis Centre, School of Physics, University of Bristol, Bristol BS8 1TL, UK
[4]Swiss Light Source, Paul Scherrer Institute, 5232 Villigen, Switzerland
[5]Cavanilles Institute of Biodiversity and Evolutionary Biology, University of Valencia, C/Catedrático José Beltrán Martínez no 2, Paterna Valencian 46980, Spain

 AA-Ö, 0000-0003-0660-3139; XW, 0000-0002-8559-7592;
PGM, 0000-0003-3395-8656; MG-S, 0000-0002-8293-3634;
MH, 0000-0001-8141-0148; FM, 0000-0002-3467-8763;
CM-P, 0000-0001-7795-5997; PCJD, 0000-0003-3116-7463

Conodont elements, microfossil remains of extinct primitive vertebrates, are commonly exploited as mineral archives of ocean chemistry, yielding fundamental insights into the palaeotemperature and chemical composition of past oceans. Geochemical assays have been traditionally focused on the so-called lamellar and white matter crown tissues; however, the porosity and crystallographic nature of the white matter and its inferred permeability are disputed, raising concerns over its suitability as a geochemical archive. Here, we constrain the characteristics of this tissue and address conflicting interpretations using ptychographic X-ray-computed tomography (PXCT), pore network analysis, synchrotron radiation X-ray tomographic microscopy (srXTM) and electron back-scatter diffraction (EBSD). PXCT and pore network analyses based on these data reveal that while

white matter is extremely porous, the pores are unconnected, rendering this tissue closed to postmortem fluid percolation. EBSD analyses demonstrate that white matter is crystalline and comprised of a single crystal typically tens of micrometres in dimensions. Combined with evidence that conodont elements grow episodically, these data suggest that white matter, which comprises the denticles of conodont elements, grows syntactically, indicating that individual crystals are time heterogeneous. Together these data provide support for the interpretation of conodont white matter as a closed geochemical system and, therefore, its utility of the conodont fossil record as a historical archive of Palaeozoic and Early Mesozoic ocean chemistry.

## 1. Introduction

Conodonts are an extinct lineage of primitive vertebrates that have been widely accepted as the earliest members of our evolutionary lineage to possess a mineralized skeleton, manifest as a complex array of 'elements' that comprise a feeding apparatus. As such, the conodont skeleton is of great significance because of the insights it provides into the biology and function of the skeleton of the most primitive vertebrates. However, the conodont skeleton harbours additional and more general insights as an archive of seawater chemistry and temperature, preserved in the calcium phosphate mineral from which the elements are composed, providing fundamental insights into past climates throughout their 300 Myr evolutionary history, including episodes of past climate change associated with mass extinctions, analogous to present [1,2].

Conodont elements are bicomponent, composed of a basal body and crown that grew in synchrony as evidenced by the growth lines that pass between these structures [3,4]. The basal body is microcrystalline and organic rich [3,5], at least in comparison to the more coarsely crystalline crown that is usually composed of a transparent lamellar crown tissue (also known as hyaline or lamellar tissue) and the enigmatic opaque white matter (also known as true white matter, albid or cancellous tissue), that may be densely to incipiently developed and variably distributed within cusps and denticles, or completely absent from conodont elements [3,6,7]. Pseudo white matter is also known from the crown of certain euconodonts; this is a distinct modified form of lamellar crown tissue [3]. White matter has been suggested as the principal tissue to target for geochemical analyses because of its assumed low permeability [6]; however, there is little agreement on the physical nature of the crown tissues (see discussion in [7] and references herein) and still less agreement on their inferred development, which is crucial for elucidating their significance in understanding the early evolution of vertebrate skeletons [8].

This disagreement is clearest in debate over the structure of white matter which has been interpreted as (i) microcrystalline [3], (ii) macrocrystalline [7], and (iii) non-crystalline [9]. Clearly, these three competing interpretations cannot all be correct and the differences in opinion stem from the fact that no study has conclusively demonstrated the structure of white matter or its relationship with the surrounding tissues; instead, they have inferred its nature on the basis of circumstantial evidence. Donoghue [3] did not resolve crystal size in white matter and so inferred it was beyond the resolution of the techniques he employed; Trotter *et al.* [7] describe crystals hundreds of micrometres in length, assuming they were single crystals based principally on the continuity of the orientation of the *c*-axis between sample points and spot electron diffraction patterns. Pérez-Huerta *et al.* [9] concluded that white matter is non-crystalline because they were not successful in obtaining diffraction patterns.

The argument supporting the sampling of the white matter for geochemistry is based on the assumption that this tissue is the least permeable of all conodont tissues and, therefore, the most resistant of bio-apatites to postmortem alteration [6]. The nature of the conodont ultrastructure (crystal structure, porosity and permeability) has significant implications for its susceptibility to diagenesis, where high permeability can facilitate the accommodation of secondary precipitates and chemical interchange during diagenesis [6]. Hence, the lack of direct evidence on the permeability of conodont white matter constitutes an important caveat on existing geochemical measurements and the inferences that have been developed from them. However, the absence of data on the porosity and permeability of conodont white matter does not occur for lack of effort. The pores mostly occur at micrometre scale and it is technically challenging to characterize them tomographically because traditional methods have lacked sufficient resolution.

Here we attempt to resolve the nature, porosity and permeability of conodont white matter using high-resolution non-invasive ptychographic X-ray-computed tomography (PXCT) and pore network analysis, combined with synchrotron radiation X-ray tomography (srXTM) and electron back-scatter diffraction

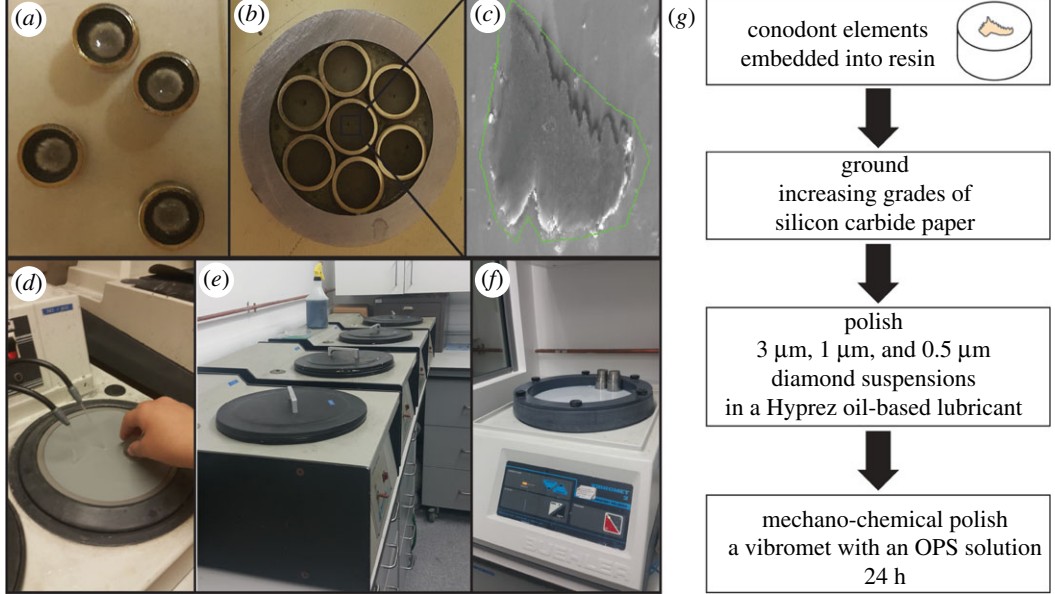

**Figure 1.** Simplified steps of the EBSD analysis, (a,b) conodont elements embedded into resin, (c) SEM picture showing the grounded surface for the EBSD analysis; (d,f) sample grinding process: (d) grinding using increasing grades of silicon carbide paper, (e) polishing using 3 μm, 1 μm and 0.5 μm diamond suspensions in a Hyprez oil-based lubricant, and (f) mechano-chemical polishing using a Vibromet with an OPS solution for 24 h; (g) schematic diagram summarizing the steps of the EBSD analysis.

(EBSD). Through analysis of taxonomically disparate Late Cambrian and Silurian conodont elements, we show that while conodont white matter is porous, these pores are not connected to one another, or to the element surface. We also demonstrate that conodont white matter is macrocrystalline, composed of crystals that are many tens of micrometres in dimensions that cross episodic growth layers. These data indicate that conodont white matter grew syntactically, indicating that individual crystals are time heterogeneous. Furthermore, our demonstration that conodont white matter exhibits low permeability provides support for the view that it represents a closed system for sampling palaeo ocean chemistry [6,7].

## 2. Material and methods

### 2.1. Materials

We investigated specimens of *Ozakodina confluens* from Prior's Frome, Late Ludfordian (Ludlow) of Worcestershire (United Kingdom), Late Silurian in age (figure 1c) and *Teridontus nakamurai* an early euconodont from the Chatsworth Limestones (Queensland, Australia), Late Cambrian in age (figure 1a), since they both exhibit well-developed white matter. Indeed, elements from both these taxa have been used in characterizing the nature of white matter [3]. *Teridontus nakamurai*, in particular, was selected because its cusps are approximately equant in cross-section and their absolute dimensions are compatible with the 100 μm diameter limitation of ptychographic nanotomography analysis. The analysed elements are well preserved with colour alteration index of 1. The specimens are reposited in the School of Life Sciences, University of Bristol, and the tomographic data are available from the University of Bristol data repository (data.bris). No ethical assessment or permission were required prior to conducting our research.

### 2.2. Ptychographic nanotomography

Conodont elements are typically in the size range of 1 mm or less, hence, attempts to characterize their structure have been challenged by difficulty of their manipulation and the scale of their substructures that require analysis. This challenge is made all the greater since conodont elements are composed largely of brittle enamel-like tissues that contain few organics [3,5,7] and so their analysis is not readily amenable to decalcification and microtomy. Thus, conodont element structure has hitherto been limited largely to traditional invasive methods such as thin-sectioning for optical microscopy, ground sections for

scanning electron microscopy (SEM), ion-milling to create single sections for TEM [3,5,7], or two-dimensional X-ray diffraction [10]. More recently, non-invasive tomography has been introduced through srXTM exploiting a high-brilliance synchrotron source to provide coherent X-ray beam for absorption [11], phase-contrast tomography [12] and ptychographic nanotomography [13,14], which is a coherent diffractive imaging technique that does away with imaging lenses and delivers three-dimensional volumes, yielding definitive data on the physical structure of conodont elements. PXCT uses a coherent, spatially confined, incident X-ray illumination across which the sample is scanned, while diffraction patterns are measured at different overlapping positions. By combining scanning, iterative image reconstruction algorithms and sample rotation, PXCT achieves quantitative electron density contrast with a resolution that can be orders of magnitude better than the size of the illumination or the scanning step, but with the limitation that the total sample size cannot exceed 100 µm in diameter. In order to resolve the finest structures and vacuities within conodont white matter and to characterize the porosity and permeability of this tissue, X-ray measurements were carried out at the cSAXS beamline (X12SA), Swiss Light Source, Paul Scherrer Institut, Switzerland [15]. The photon energy was 6.2 keV. Specimens with appropriate diameter were glued directly to the top of a copper tomography sample pin [16]. Ptychographic scans were performed with a field of view of $62 \times 7 \ \mu m^2$. An Eiger photon-counting detector, positioned 7.4 m downstream of the sample, was used to acquire the far-field intensity patterns at each of the resulting 392 scanning points, each with an exposure time of 200 ms [17]. The same ptychography scan was repeated at 400 equiangular orientations of the sample in the range between 0 and 179.5°. The ptychography reconstructions were carried out using the difference-map algorithm [18] followed by a maximum-likelihood refinement [19] resulting in a reconstruction pixel size of 46.8 nm. These two-dimensional projections were further processed and aligned using in-house scripts [20]. A half-period resolution of 150 nm was determined using Fourier shell correlation with the half-bit criterion [21].

## 2.3. Synchrotron radiation X-ray tomographic microscopy

Measurement with larger field of view of the same specimens was obtained using srXTM at X02DA TOMCAT beamline of the Swiss Light Source, Paul Scherrer Institute. We used a 40× objective lens, energy of 17 keV and with an exposure time of 300 ms, acquiring 1501 projections equiangularly over 180°. The ensuing projections were post-processed and rearranged into flat- and dark-field-corrected sinograms, and reconstruction was performed on a 60-core Linux PC farm using a Fourier transform routine and a regridding procedure [22]. The resulting data achieved a voxel size of 0.1625 µm. A half-period resolution of 0.325 µm was determined by inspection of an edge response.

## 2.4. Computed tomography and pore network analysis

The reconstructed tomographic data from PXTM and srXTM were analysed using AVIZO LITE v. 9.1 (Thermofisher), allowing us to extract quantitative three-dimensional digital models and virtual thin sections. Pore Network Analysis was performed with the Avizo X Pore Network Modeling extension to constrain the volume fraction of connected porosity versus isolated pore space. This allowed us to measure the number, size and distribution of the pores and the degree of connectivity, as a proxy of the material permeability [23,24]. The volume of an individual pore space was approximated based on the radius (EqRadius) of a sphere that can fit within it.

## 2.5. Electron back-scatter diffraction

Sample preparation for EBSD is so critically important that any surface deformation or contamination can prevent diffraction pattern collection. Hence, a diversity of sample preparation techniques are available for EBSD based on the structure and composition of a sample. Despite this, there has been little detailed assessment of sample preparation for EBSD analyses outside of biogenic carbonates [9,25,26]. We experimented with a diversity of preparation techniques that have been applied for EBSD analysis. Conodont elements were embedded into different types of resin including non-conductive araldite epoxy resin, hot mounting PolyFast containing graphite powder and conductive silver epoxy. In order to obtain good results from EBSD analysis, the analysed samples must be free of damage to the crystal lattices at the surface; therefore, the polishing of the specimens is critically important. Initially, the prepared samples were sequentially ground using increasing grades of silicon carbide paper before polishing using 3 µm, 1 µm and 0.5 µm diamond suspensions in a

Hyprez oil-based lubricant. Mechanical grinding and polishing of the samples easily damage the crystal lattices near the surface of the studied materials. Therefore, it was necessary to perform additional chemical polishing of the samples. Hence, after grinding and polishing of the samples, we employed a mechano-chemical polish using a vibromet for most of the samples. We also explored the utility of a low energy SEM-hosted focused ion beam etch of a small region of the specimen to achieve an optimally planar and polished surface. Despite our best efforts, no crystallographic information, diffraction patterns or EBSD maps could be obtained. Ultimately, our only successful conodont element preparations for EBSD were achieved through embedding into non-conductive araldite epoxy resin and grinding in sequential steps using silicon carbide papers, polishing using 3 µm, 1 µm and 0.5 µm diamond suspensions in a Hyprez oil-based lubricant and finally a 24 h mechano-chemical polish using a vibromet with an OPS solution (figure 1). Some samples were also carbon coated or surrounded by silver paint after embedding in non-conductive epoxy resin in order to improve their conductivity.

EBSD analyses were performed on a Zeiss™ SIGMA™ Variable Pressure (VP) SEM, operated in the instruments low-vacuum (VP) mode to negate against the deleterious impacts of sample charging under the incident electron beam. Using the system's high-current mode, an accelerating voltage of 20 kV, 120 µm aperture and 2.4 nA beam current were used to generate the diffraction signal from the surface of the specimen. An EDAX (Ametek Inc.) DigiView-4 high-speed camera with associated orientation image mapping (OIM) data collection (V. 7) beam control, Kikuchi pattern matching and visualization software was used to map the crystallographic nature of the sample, employing step sizes of 500 nm–2 µm and 4 × 4 data binning over the predetermined region of interest. The accompanying OIM analysis (V. 7) package was subsequently used to generate the crystal orientation maps and perform statistical analysis of the EBSD data; including the production of inverse pole figure maps and plots, through the phase data selected in order to undertake the earlier EBSD mapping acquisition.

# 3. Results

## 3.1. srXTM and ptychographic nanotomography

The srXTM characterization of a *Teriodontus nakamurai* element allowed us to resolve the histology of the crown. An X-ray homogeneous lamellar crown tissue is present at the base of the crown and the lower margins of the cusp, whereas more heterogeneous white matter comprises the bulk of the cusp itself. This tissue is distinguished principally on the basis of small void spaces that vary in density along the long axis of the cusp (figure 2*b*). However, even at 0.325 µm resolution, it is still not possible to discern whether the void spaces are interconnected. PXCT characterization of a portion of the cusp provides 150 nm resolution which reveals that the void spaces are arranged in approximately concentric circles paralleling the surface circumference of the cross-section of the cusp (figure 2*f*–*g*). The largest void spaces are sited at the centre, proximal to which the concentric arrangement voids is most coherent, decreasing in size centrifugally, showing a wide range of void sizes (with several orders of magnitude; see electronic supplementary material, table S1 and figure 2*g,i*).

## 3.2. Analysis of the pore network

Pore network analysis of the PXCT data allowed us to characterize the porosity of a portion of the *Teridontus nakamurai* cusp in terms of pore numbers, size, distribution and connectivity. From an extracted slice (figure 2*c*), with a volume of 14 593.4 µm³, a total of 10 898 voids were identified and measured (5051 voids excluding pores smaller than 150 nm diameter due the scan resolution, representing 0.5% of the total volume). The pores exhibit a wide range of volumes, 0.001787–1.137 µm³, with a mode of 0.166866779 µm Eqradius (figure 2*h*). In addition, pore network analysis allowed us to visualize the porosity network and discern between the connected porosity and the isolated pore space, visualizing those void areas (pores) as spheres and their interconnections (throats) as lines. As the three-dimensional visualization demonstrates (figure 2*g*), linear throats cannot be seen (supported by the throat analysis, see electronic supplementary material, table S1), suggesting that the total porosity of our sample, independent of their size and distribution, are neither interconnected nor connected the element surface, indicating that despite its porosity, white matter is a closed system.

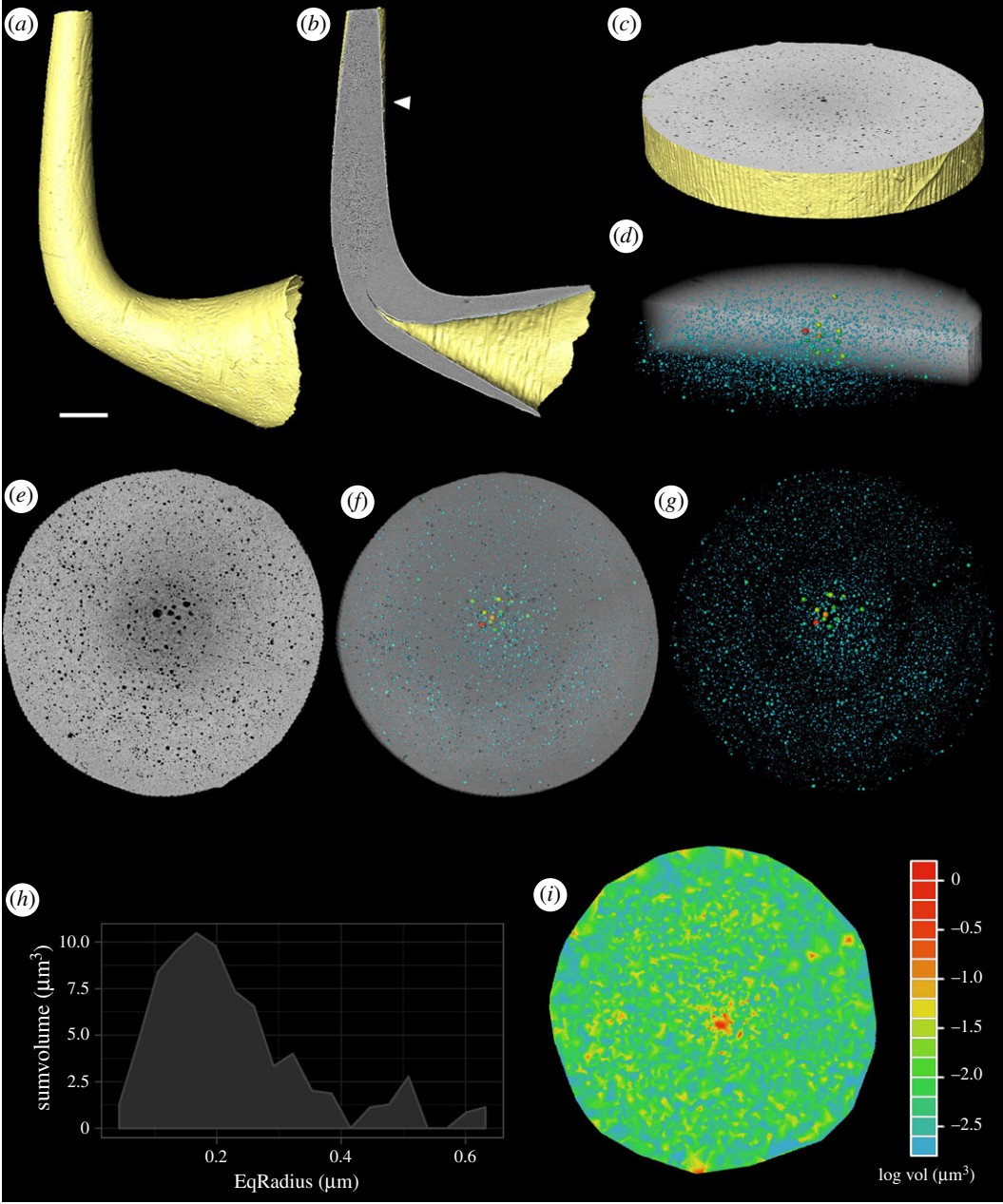

**Figure 2.** Tomographic and pore network analyses of the euconodont *Teridontus nakamurai*. (*a,b*) Surface rendering derived from the srXTM data, (*a*) lateral view and (*b*) 'virtual thin sections' clearly resolving the histology of the crown; (*c*) surface rendering and (*e*) 'virtual thin sections' from the same specimen obtained from a single scan at the middle of the cusp using PXCT; (*d,f,g*) extracted topological porosity and networks derived from the PXCT data showing the largest void spaces (spheres) sited at the centre, decreasing in size centrifugally, and not connected between them (throats); (*h*) pore size (volume) distribution of the volume analysed, showing a pick of the most abundant pore size around 0.16 μm EqRadius (excluding pores smaller than 150 nm); (*i*) colour heat map derived from the pore network analysis plotting the pore distribution by size (excluding pores smaller than 150 nm) at the cusp cross section, confirming the visual analysis that the largest pores are concentrated at the center of the cusp. Scale bar for (*a,b*) 62 μm; (*c–g*), (*i*) 10 μm.

## 3.3. Electron back-scatter diffraction

We were able to obtain coherent diffraction patterns for white matter in the denticles of elements of both *Teriodontus nakamurai* and *Ozarkodina confluens*, but not the surrounding lamellar crown tissue. In both instances, these diffraction patterns reveal that the white matter comprising the cores and tips of denticles occurs in each instance as a single crystal (figure 3*b,e,f*). There is a strong preferred crystallographic orientation in relation to the whole conodont element. The white matter cores of

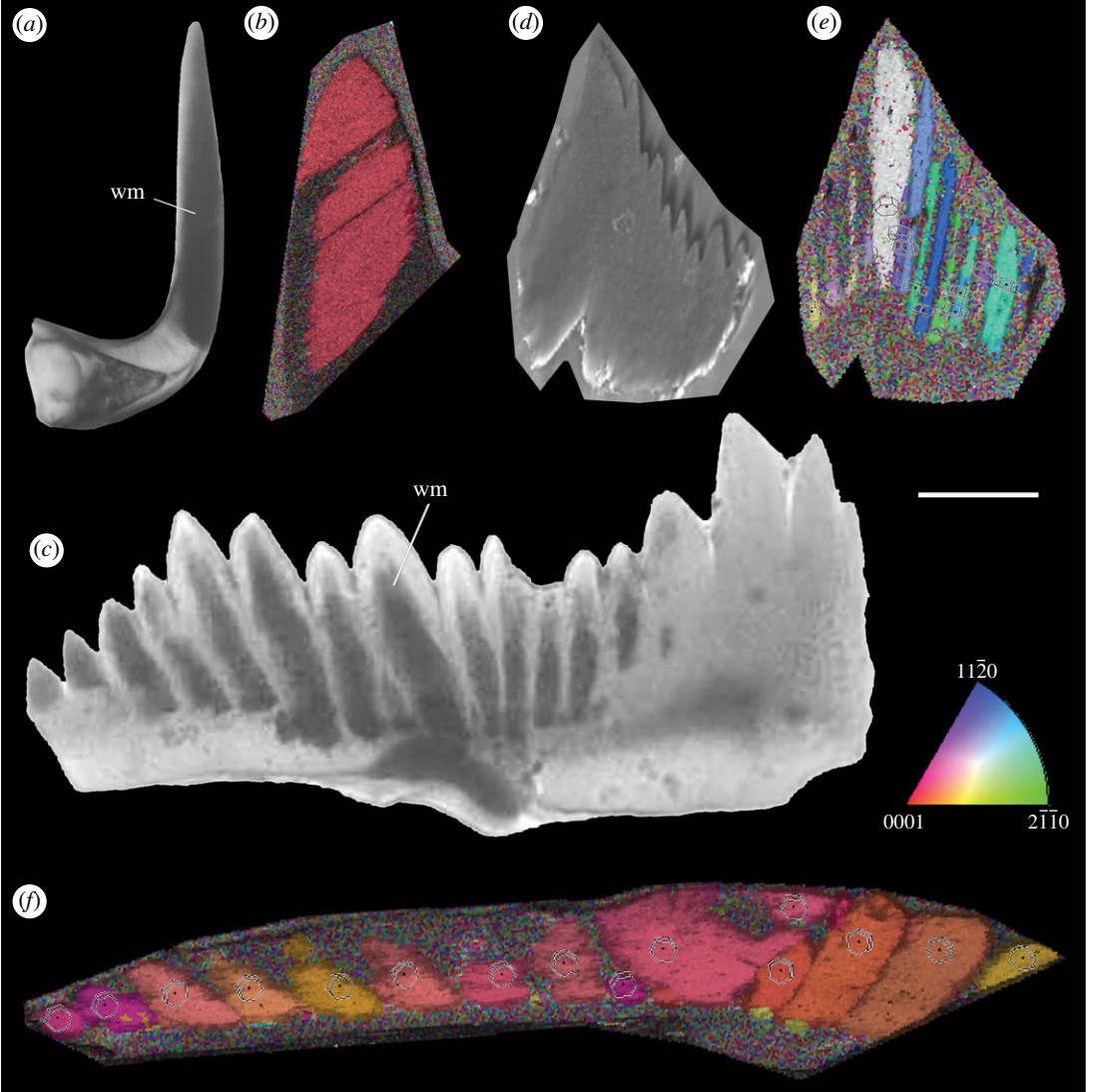

**Figure 3.** EBSD analysis of the euconodont *Teridontus nakamurai (a)* and *Ozakodina confluens* (*c*) where the white matter (dark areas with transmitted light) at the cores of the main cusp and denticles are clearly discernible; (*b*) longitudinal cross section (transversal plane) showing the crystal orientation map of *Teridontus* and the *c*-axis of the crystal that comprises the white matter aligned with the long axis of the cusp; (*d*) *Ozarkodina* conodont element embedded into resin for the EBSD analysis; (*e,f*) longitudinal cross section (transversal plane) showing the crystal orientation maps of *Ozakodina* according to the crystallographic colour key with the same orientation showed in (*b*). As seen in *Teridontus,* the white matter of the cores and tips of denticles behave as single crystals showing similar orientations to adjacent denticles. Crystallographic key indicating colour coding of crystallographic planes. Scale bar for (*a*), 130 µm, (*b*) 45 µm, (*c–e*) 100 µm and (*f,g*) 60 µm. Abbreviations: white matter (wm).

adjacent crystals exhibit similar orientations, with a gradual overall change in crystal orientation among denticles from the extreme of one process to another (e.g. figure 3*e,f*). The *c*-axes of the crystals comprising the white matter are essentially aligned with the long axis of each denticle, approximately parallel to the local upper–lower axis of individual elements. The microstructure of the conodont elements exhibited an alignment principally in the [001] direction.

## 4. Discussion

### 4.1. The crystalline nature of conodont white matter and implications for histogenesis

The results of our EBSD characterization of conodont element white matter demonstrate unequivocally that this tissue is crystalline and that the white matter comprising each denticle core is composed of a

single crystal at the scale of the denticle itself. This result contradicts Donoghue [3], who interpreted white matter as nanocrystalline, principally on the basis of TEM analyses by Pietzner et al. [5] and his failure to resolve crystal dimensions using scanning electron microscopy. However, our results substantiate the conclusions of Trotter et al. [7], that white matter is macrocrystalline, based on more limited observations. We reject the conclusion of Pérez-Huerta et al. [9] that conodont elements in general, and white matter in particular, are non-crystalline. We attribute the failure of Pérez-Huerta et al. [9] to obtain diffraction patterns to the extreme difficulty of preparing conodont elements in particular, and biophosphates in general, for EBSD analysis.

The macroscopic scale of the white matter crystals is potentially incompatible with the evidence of episodic growth of the elements themselves [27,28] since the white matter denticle cores, which equate to the individual crystals observed here, transcend the scale of growth stages. However, unequivocal evidence of episodic growth in conodont elements [27,28] indicates that while lamellar crown tissue is appositional, white matter growth is syntactic, growing in crystallographic continuity with the white matter deposited in previous growth stages. This does not preclude the growth of white matter in step with the lamellae of crown tissue, for which there is clear evidence [3] and, indeed, the arrangement of pores within the white matter likely reflects its periodic growth within growth episodes.

## 4.2. The porosity, permeability and functional adaptation of conodont white matter

The results of our pore network analyses of the PXCT data demonstrate that conodont white matter is porous but not permeable (to within a lower limit of 150 nm) as previously concluded by Trotter et al. [6,7]. The void spaces are isolated and arranged in concentric cylinders that parallel the external surface of the element. The concentric arrangement of the voids is particularly clear close to the core of the white matter, resembling the appearance of growth lines that are known to occur in white matter [3]. The voids are generally larger at the core (figure 3g,h) and, as such, it may be tempting to interpret this phenomenon as reflecting the coalescence of voids during the diagenetic transformation of lamellar crown tissue white matter [29]. However, white matter is bounded by growth layers within lamellar crown tissues, evidencing its primary biological, and not diagenetic origin [3]. Thus, it is more likely that the larger voids at the core of the white matter simply reflect topological differences in the dimensions of voids, between the tip and flanks of denticles. This is because the core of a block of white matter represents the tip of a denticle at an earlier stage of development, and it is at the tip of the denticle that mineral reorganization is most likely to occur when new growth is initiated. Moving from the core to the margins of a block of white matter, the voids are increasingly restricted to concentric zones, presumably reflecting growth.

The macrocrystalline nature of conodont white matter is surprising but likely represents a functional adaptation to the sophisticated tooth functions of conodont elements [30,31]. This is especially challenging given the otherwise brittle nature of lamellar crown tissue (which exhibits widespread evidence of functionally induced spalling as a consequence of element–element interaction [31,32]) and the proportionally high forces induced in these microscopic dental tools [33,34]. Jones et al. [33,34] have shown through finite element modelling that the differentiation of euconodont element crowns into lamellar crown tissue and white matter allowed cusps and denticles to withstand greater tensile stresses than cusps comprised solely of lamellar crown tissue. Mineralogically and crystallographically homogeneous, non-porous macrocrystalline white matter might be vulnerable to crack propagation. However, the occurrence of inhomogeneities in the form of small equant void spaces within white matter would have served to decuss propagating cracks [33,35]. This occurs because the energy required to propagate cracks scales with the radius of the crack tip [36]. Thus, we conclude that the porosity of white matter is an adaptation to prevent crack propagation. The low permeability of white matter is probably also functionally important as it might otherwise serve to guide crack propagation.

## 4.3. Implications for the utility of conodont elements as archives of palaeo ocean chemistry

Above all, our results are significant for the exploitation of conodont biomineral as an archive of palaeo ocean chemistry. We evidence the conclusions of Trotter et al. [6,7] that conodont element white matter is a suitable archive since though it is porous, it has very low permeability not merely because the void spaces are generally unconnected (although we cannot formally exclude the possibility that connections occur below the 150 nm resolution of the PXCT measurements), but also because they are hosted within a macrocrystalline tissue with diminishing permeability effected by crystal boundaries. This is important, since it renders conodont white matter an effectively closed system, protected from

the effects of diagenetic alteration. That said, conodont samples often exhibit postmortem alteration, including breakage, abrasion and tectonic shearing, though only the latter is likely to increase permeability and diminish the quality of samples for geochemical analysis. The impact of thermal alteration on the nanoscale character of white matter is not known, but it is most likely to occlude pores and diminish permeability. Nevertheless, it is clear that samples should be chosen judiciously for tissue-specific geochemical analysis.

Traditional oxygen isotope analyses comprise the most common geochemical studies of conodont apatite by using conventional wet chemistry techniques [37–39], but do not have the ability to discriminate different histological components. However, the spatial scale of conodont white matter crystals, effectively equivalent to the dimensions of whole cusps and denticles, renders feasible tissue-specific analysis using *in situ* techniques, such as ion microprobes [2,28,40–42]. Given the syntactic episodic growth of white matter crystals through the lifetime of an individual, we should anticipate disparate but clustered measurements along the base-crown axis of an individual cusp or denticle, potentially providing insights into secular changes in sea water chemistry during conodont element growth.

## 5. Conclusion

This study aims to resolve a physical characterization of conodont white matter structure using state-of-the-art equipment and methods and to inform their utility as geochemical archives of past ocean chemistry, corroborating previous conclusions [6,7]. The EBSD analyses of conodont elements reveal that the conodont white matter having important consequences for investigation of past climate changes is macrocrystalline crossing episodic growth layers. Additionally, the PXCT and pore network analyses characterized this tissue as porous but not permeable, since pores are not connected to each other and with the exterior and are arranged in concentric cylinders that parallel the external surface of the element. These data are compatible with white matter growing syntactically in crystallographic continuity with the white matter deposited in previous growth stages. Consequently, our results emphasize the significance of conodont elements providing insights into changes in ocean chemistry through geologic time due to its microcrystalline, porous and low permeable characteristics.

Data accessibility. The original tomographic raw data, as well as the Avizo projects with the segmented information and the Pore Network Analysis (see readme file) are hosted at the data.bris Research Data Repository (doi:10.5523/bris.3rjh9x2tbviqs2j6sbuuren8l1).

Authors' contributions. A.A.-O., C.M.P. and P.C.J.D. designed the study. A.A.-O., X.W. and P.G.M. carried out the EBSD analysis; M.G.-S., M.H. and C.M.P. scanned the specimens at the cSAXS, and F.M. scanned the samples at the TOMCAT, both at the Paul Scherrer Institute; A.A.-O., C.M.P. and P.C.J.D. wrote the manuscript, with critical revision from all authors. All the authors gave final approval for submission.

Competing interests. We declare that we have no competing interests.

Funding. C.M.P. and P.C.J.D. were supported by Paul Scherrer Institute to conduct the cSAXS experiments and by the Ministry of Science and Innovation of Spain, Research Project PID2020-117373GA-I00.

Acknowledgements. We acknowledge the Paul Scherrer Institute, Villigen, Switzerland for provision of synchrotron radiation beamtime at the TOMCAT and cSAXS beamline of the SLS. We thank Robert S. Nicoll for donating the samples of *Teridontus nakamurai* used in this analysis; Sarawuth Wantha (thermofisher) for his technical support to develop the Pore Network Analysis; and three anonymous reviewers whose comments improved the final version of the manuscript.

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
