## [Peer Review File · Royal Society Open Science]

Review History

RSOS-202013.R0 (Original submission)

Review form: Reviewer 1 (Duncan Murdock)

Is the manuscript scientifically sound in its present form?

Yes

Are the interpretations and conclusions justified by the results?

Yes

Is the language acceptable?

Yes

Do you have any ethical concerns with this paper?

No

Have you any concerns about statistical analyses in this paper?

No

Recommendation?

Accept with minor revision (please list in comments)

Comments to the Author(s)

This is an excellent piece of work, both in terms of the technical challenges that have been overcome and the presentation of the results. It informs both the palaeobiology of an important group, and the utility of conodonts for paleoenvironmental analyses.

I think the authors could expand their discussion a little in two ways, but otherwise it is well written and comprehensive. Firstly, these findings have implications for the mechanisms for growth of conodont elements, which is still one of the big outstanding questions for this group. Could the authors provide any insight into the biological control of simultaneous production of lamellar crown and syntactical crystalline white matter? Particularly in tissue with such a low organic component. Secondly, it is common to see post-mortem breakage, abrasion, and/or tectonic shearing in conodont elements (even with low CAI), could the authors comment on the potential likely effects of these on the porosity and permeability, and thus inform best practice for selecting specimens for analysis?

In addition, I have the following minor corrections:

Two sentences in the abstract are so concise they are incomplete:

P1. L23. "yielding fundamental insights into the palaeotemperature and chemical composition" of what?

P1. L29. "reveal that while white matter is extremely porous and the pores are unconnected.". so...?

P6. L182-186. This long sentence is difficult to unpick.

Figure 2. Can you indicate the orientations of the cross sections in b, d and e.

Review form: Reviewer 2 (Julie Trotter)

Is the manuscript scientifically sound in its present form?

Yes

Are the interpretations and conclusions justified by the results?

Yes

Is the language acceptable?

Yes

Do you have any ethical concerns with this paper?

No

Have you any concerns about statistical analyses in this paper?

No

Recommendation?

Accept with minor revision (please list in comments)

Comments to the Author(s)

The authors present a thorough, quantitative study of the crystalline nature of conodont crown tissues using the benefits of present-day high-resolution technologies. The combination of techniques used here provides indisputable proof of the large, single crystal nature of conodont albid tissue (aka white matter), its low permeability despite its high porosity, and its primary biological origin, which confirm the conclusions of earlier studies that are often ignored by researchers. These data have important implications for both understanding the growth of conodont mineralised tissues, and the suitability of albid tissue for geochemical analysis for palaeoenvironmental reconstructions.

This paper is worthy of publication in *RSOS*, however several corrections should be made (see below) and I would hope to see a slightly more nuanced assessment of prior work. This study unequivocally confirms the observations reported and conclusions drawn by Trotter and co-workers some 14 years ago. Convincingly, albeit arguably not totally conclusive, Trotter et al (2007) reported the single crystal and non-permeable yet highly porous nature of albid tissue (ie. the 2 key conclusions of the present study) from combined extinction contours and TEM spot diffraction patterns, as well as reconnaissance EBSD results, in a range of conodont species. Those data were combined with a thorough geochemical study (Trotter et al 2006) to support those authors' conclusions regarding the primary biological origin and suitability of albid tissue as an environmental archive (the 3rd conclusion of the present study), which should not simply be dismissed as "without material evidence". Additional citations in the text are also warranted.

Specific comments:

The introduction should canvas the complexities of both nomenclature and variability of conodont histologies, eg. cancellate ('true') albid crown that differs from apparent white matter (voids) in lamellar crown which have been previously discussed thoroughly by Donoghue 1998 and Trotter et al 2007. These issues lead to inconsistencies and confusion when describing and identifying crown tissues by various researchers, including in geochemistry studies. Note that line 225 in this paper has the first and only mention of the term "albid". It is important to clarify these terms and acknowledge differences, citing relevant studies for further reading. Despite the outcomes presented in earlier studies by Trotter and her colleagues, consistent with the findings of this recent submission, geochemistry studies sometimes contradict and even erroneously report those earlier results. These new data are therefore a very welcome contribution that will likely influence future conodont work, hopefully across all disciplines. It should also be noted that, given the widespread use of lamellar tissue in geochemistry studies (see below), better constraints on its permeability beyond those discerned in previous studies would also be helpful.

Line 24-25:

It is the high porosity of white matter/albid crown from its well-known 'cancellate' nature that has led many researchers to believe it is permeable (note the important distinction between porosity and permeability), despite contrary evidence and conclusions reported by Trotter and her colleagues.

Please revise the following excerpt: "...however, the crystallographic nature of the white matter and its inferred permeable structure, may make it unsuitable as a geochemical archive." change to "...however, the porosity and crystallographic nature of the white matter and its inferred permeability is disputed, which questions its suitability as a geochemical archive."

Line 26:

"We attempted to discriminate among these competing interpretations...". Revise this phrase as it doesn't really make sense and the competing interpretations have not actually been stated or described (hence changes to line 25 above), so best to describe what you are actually doing: eg.

“We attempted to better constrain these characteristics of white matter and address conflicting interpretations...”

Lines 53-55:

“enigmatic white matter that is the principal tissue targeted for geochemical analyses because of its assumed low permeability [6]”.

I suggested that albid cancellate specifically (as opposed to lamellar ‘white matter’) is the preferred histology to target for geochemical studies, but this is rarely followed. Thus, the statement by these authors is completely incorrect so needs to be revised, eg. end the sentence at “...enigmatic white matter.” Few conodont geochemical palaeoseawater reconstruction studies specifically target white matter for various reasons:

1. Oxygen isotope analyses comprise the most common geochemical studies of conodont apatite. These have mostly been bulk analyses using conventional wet chemistry techniques, which do not have the ability to discriminate different histological components, hence include both basal material and all crown histologies. Following the development of in-situ ion microprobe $\delta^{18}\text{O}$ conodont analyses (Trotter et al 2008), which enables specific histologies to be targeted, new studies are being undertaken yet none to date have specifically only targeted white tissue for palaeoseawater reconstructions. Both histologies are targeted in my studies, to allow ongoing comparison of their compositions (which have shown no measurable systematic difference), whereas ‘hyaline’ (lamellar tissue) is most commonly analysed by other researchers given its smooth surface which is (erroneously) believed to be better for ion probe analyses.
2. There are few laser ablation studies, since the initial work of Trotter et al 2006, which can be used for in situ trace element (LA-ICPMS) or some isotope analyses (LA-MC-ICPMS), and again do not specifically target white tissue mostly due to the spatial resolution and precision required. For REE studies, for example, hyaline tissue can only be used given the low to absent concentrations of REE in albid tissue, which is also the case for many other trace elements (and/or isotope studies).

Line 93:

Add citations “...support for the view that it represents a closed system for sampling past palaeo ocean chemistry (Trotter et al 2006, 2007)”.

Lines 228-229:

“...substantiate the view of Trotter et al...” please replace “view” with “conclusions”
 “...which they inferred by did not demonstrate” ...please replace with “based on more limited observations”. Those earlier studies were not baseless or lacking evidence, as you seem to imply.
 “We can nevertheless..” should be “We can certainly..”

Line 244:

Add citations “conodont white matter is porous but not permeable (to within a lower limit of 150 nm).” ie. as previously concluded by Trotter et al (2006, 2007).

Line 277:

Add 2006 reference: “We evidence the view of Trotter and colleagues [6, 8]...”

Line 287:

“...chemistry over interannual timescales.” I advise that “interannual” is not accurate. We are unlikely to ever (unequivocally) know the longevity of conodonts and hence the time span that their elements represent, and presently, few instruments (if any) could provide useful data at such high-spatial resolution from those specific zones, ie. at sufficient precision within analytical uncertainties. Furthermore, we cannot rule out ontogenetic changes, which commonly occur in biominerals that also depends on the chemical element (or isotopic system) being analysed.

I would therefore suggest revising this sentence to “potentially providing insights into secular changes in sea water chemistry during conodont element formation.”

Supplementary file:

Note that the supplementary table is not cited in the text. In that file, I also suggest to add one text line in the empty sheet, “throat”, stating “no throat connections discerned”.

Figures:

Can any high power images (eg. SEM) of sections through the histologies analysed be included? Such images, which are most commonly used and seen by researchers, both conodont workers and geochemists, when describing and selecting samples for analysis would be useful to compare with the images presented. This is also relevant given the complexities and confusion in recognising various types of white matter and lamellar histologies, as mentioned above.

Review form: Reviewer 3 (Alberto Perez-Huerta)

Is the manuscript scientifically sound in its present form?

Yes

Are the interpretations and conclusions justified by the results?

No

Is the language acceptable?

Yes

Do you have any ethical concerns with this paper?

No

Have you any concerns about statistical analyses in this paper?

No

Recommendation?

Major revision is needed (please make suggestions in comments)

Comments to the Author(s)

Authors present a very interesting useful study, and potentially impactful for the general use of conodonts as geochemical bioarchives. In particular, I commend authors for obtaining remarkable data with challenging techniques. The main drawback of the manuscript is that authors draw major conclusions about conodont element permeability and crystallinity just based on partial data of two conodont elements, which are rarely used in geochemical studies. Also, some of the results are not very well explained, in particular for EBSD. I would recommend authors to address the following comments/questions:

- The general readership may not be aware of the distribution of different tissues in the two analyzed conodont elements, so it would be useful to present a general diagram with this information.
- The porosity, and subsequent permeability discussion, is only based on data from the early euconodont *Teridontus*. Do authors have any data for *Ozakodina*? For *Teridontus*, it is stated that linear throats are not visible, and the assumption is that pores are not connected. However, the section is only at the center of the element. Would a section close to the cusp or at

the bottom of the element show something different? Could these linear throats present but below the technique resolution (i.e., < 100 nm)?

- Although PXCT and synchrotron tomography are not usually accessible to many researchers, EBSD is becoming a routine technique in many microscopy labs. Authors have achieved an unprecedented resolution of EBSD data for conodonts. In order to facilitate reproducibility, and wider use of this approach to the community, authors have to provide more details about the EBSD protocol under methodology (even including diagrams). Authors only state that data was achieved after 24-hour mechano-chemical polish using an OSP solution.
- The EBSD data (in Figure 2) is of very difficult interpretation. In particular, Fig. 2e is not clear what corresponds to. I would recommend showing the polished surfaces, and their correspondence to the location in the conodont element, and then the EBSD crystallographic maps matching the regions in the polishing surfaces. EBSD data without location context is not useful. Also, more interpretation of the EBSD data in terms of microstructure and element growth would be important for a better understanding of the biomineralization of conodont elements.

Decision letter (RSOS-202013.R0)

Dear Dr Martinez Perez

The Editors assigned to your paper RSOS-202013 "X-Ray nanotomography and Electron Backscatter Diffraction demonstrate the crystalline, heterogeneous and impermeable nature of conodont white matter" have now received comments from reviewers and would like you to revise the paper in accordance with the reviewer comments and any comments from the Editors. Please note this decision does not guarantee eventual acceptance.

We invite you to respond to the comments supplied below and revise your manuscript. Below the referees' and Editors' comments (where applicable) we provide additional requirements. All referees make important comments, but please pay particular attention to those of Referee 3. Final acceptance of your manuscript is dependent on these requirements being met. We provide guidance below to help you prepare your revision.

Please submit your revised manuscript and required files (see below) no later than 21 days from today's (ie 10-Feb-2021) date. Note: the ScholarOne system will 'lock' if submission of the revision is attempted 21 or more days after the deadline. If you do not think you will be able to meet this deadline please contact the editorial office immediately.

Please note article processing charges apply to papers accepted for publication in Royal Society Open Science (<https://royalsocietypublishing.org/rsos/charges>). Charges will also apply to papers transferred to the journal from other Royal Society Publishing journals, as well as papers submitted as part of our collaboration with the Royal Society of Chemistry

(<https://royalsocietypublishing.org/rsos/chemistry>). Fee waivers are available but must be requested when you submit your revision (<https://royalsocietypublishing.org/rsos/waivers>).

Kind regards,

Anita Kristiansen
Editorial Coordinator

on behalf of Professor Elizabeth Harper (Associate Editor) and Peter Haynes (Subject Editor)
openscience@royalsociety.org

Associate Editor Comments to Author (Professor Elizabeth Harper):

Comments to the Author:

This is potentially a very interesting study. It is technically difficult and the results are significant advances. All three reviewers bring up relevant points which should all be addressed. In particular the need for:

- better description of EBSD preparation such that the findings might be reproducible
- inclusion of Ozarkodina data for all methods not just EBSD
- appropriate recognition of previous work

It would be interesting if the findings could be, even if briefly, considered as sources of further or new information about the mechanisms of element biomineralization.

Reviewer comments to Author:

Reviewer: 1

Comments to the Author(s)

This is an excellent piece of work, both in terms of the technical challenges that have been overcome and the presentation of the results. It informs both the palaeobiology of an important group, and the utility of conodonts for paleoenvironmental analyses.

I think the authors could expand their discussion a little in two ways, but otherwise it is well written and comprehensive. Firstly, these findings have implications for the mechanisms for growth of conodont elements, which is still one of the big outstanding questions for this group. Could the authors provide any insight into the biological control of simultaneous production of lamellar crown and syntactical crystalline white matter? Particularly in tissue with such a low organic component. Secondly, it is common to see post-mortem breakage, abrasion, and/or tectonic shearing in conodont elements (even with low CAI), could the authors comment on the potential likely effects of these on the porosity and permeability, and thus inform best practice for selecting specimens for analysis?

In addition, I have the following minor corrections:

Two sentences in the abstract are so concise they are incomplete:

P1. L23. "yielding fundamental insights into the palaeotemperature and chemical composition" of what?

P1. L29. "reveal that while white matter is extremely porous and the pores are unconnected." so...?

P6. L182-186. This long sentence is difficult to unpick.

Figure 2. Can you indicate the orientations of the cross sections in b, d and e.

Reviewer: 2

Comments to the Author(s)

The authors present a thorough, quantitative study of the crystalline nature of conodont crown tissues using the benefits of present-day high-resolution technologies. The combination of techniques used here provides indisputable proof of the large, single crystal nature of conodont albid tissue (aka white matter), its low permeability despite its high porosity, and its primary biological origin, which confirm the conclusions of earlier studies that are often ignored by researchers. These data have important implications for both understanding the growth of conodont mineralised tissues, and the suitability of albid tissue for geochemical analysis for palaeoenvironmental reconstructions.

This paper is worthy of publication in *RSOS*, however several corrections should be made (see below) and I would hope to see a slightly more nuanced assessment of prior work. This study unequivocally confirms the observations reported and conclusions drawn by Trotter and co-workers some 14 years ago. Convincingly, albeit arguably not totally conclusive, Trotter et al (2007) reported the single crystal and non-permeable yet highly porous nature of albid tissue (ie. the 2 key conclusions of the present study) from combined extinction contours and TEM spot diffraction patterns, as well as reconnaissance EBSD results, in a range of conodont species. Those data were combined with a thorough geochemical study (Trotter et al 2006) to support those authors' conclusions regarding the primary biological origin and suitability of albid tissue as an environmental archive (the 3rd conclusion of the present study), which should not simply be dismissed as "without material evidence". Additional citations in the text are also warranted.

Specific comments:

The introduction should canvas the complexities of both nomenclature and variability of conodont histologies, eg. cancellate ('true') albid crown that differs from apparent white matter (voids) in lamellar crown which have been previously discussed thoroughly by Donoghue 1998 and Trotter et al 2007. These issues lead to inconsistencies and confusion when describing and identifying crown tissues by various researchers, including in geochemistry studies. Note that line 225 in this paper has the first and only mention of the term "albid". It is important to clarify these terms and acknowledge differences, citing relevant studies for further reading. Despite the outcomes presented in earlier studies by Trotter and her colleagues, consistent with the findings of this recent submission, geochemistry studies sometimes contradict and even erroneously report those earlier results. These new data are therefore a very welcome contribution that will likely influence future conodont work, hopefully across all disciplines. It should also be noted that, given the widespread use of lamellar tissue in geochemistry studies (see below), better constraints on its permeability beyond those discerned in previous studies would also be helpful.

Line 24-25:

It is the high porosity of white matter/albid crown from its well-known 'cancellate' nature that has led many researchers to believe it is permeable (note the important distinction between porosity and permeability), despite contrary evidence and conclusions reported by Trotter and her colleagues.

Please revise the following excerpt: "...however, the crystallographic nature of the white matter and its inferred permeable structure, may make it unsuitable as a geochemical archive." change to "...however, the porosity and crystallographic nature of the white matter and its inferred permeability is disputed, which questions its suitability as a geochemical archive."

Line 26:

"We attempted to discriminate among these competing interpretations...". Revise this phrase as it doesn't really make sense and the competing interpretations have not actually been stated or described (hence changes to line 25 above), so best to describe what you are actually doing: eg. "We attempted to better constrain these characteristics of white matter and address conflicting interpretations..."

Lines 53-55:

"enigmatic white matter that is the principal tissue targeted for geochemical analyses because of its assumed low permeability [6]".

I suggested that albid cancellate specifically (as opposed to lamellar 'white matter') is the preferred histology to target for geochemical studies, but this is rarely followed. Thus, the statement by these authors is completely incorrect so needs to be revised, eg. end the sentence at "...enigmatic white matter." Few conodont geochemical palaeoseawater reconstruction studies specifically target white matter for various reasons:

1. Oxygen isotope analyses comprise the most common geochemical studies of conodont apatite. These have mostly been bulk analyses using conventional wet chemistry techniques, which do not have the ability to discriminate different histological components, hence include both basal material and all crown histologies. Following the development of in-situ ion microprobe d18O conodont analyses (Trotter et al 2008), which enables specific histologies to be targeted, new studies are being undertaken yet none to date have specifically only targeted white tissue for palaeoseawater reconstructions. Both histologies are targeted in my studies, to allow ongoing comparison of their compositions (which have shown no measurable systematic difference), whereas 'hyaline' (lamellar tissue) is most commonly analysed by other researchers given its smooth surface which is (erroneously) believed to be better for ion probe analyses.
2. There are few laser ablation studies, since the initial work of Trotter et al 2006, which can be used for in situ trace element (LA-ICPMS) or some isotope analyses (LA-MC-ICPMS), and again do not specifically target white tissue mostly due to the spatial resolution and precision required. For REE studies, for example, hyaline tissue can only be used given the low to absent concentrations of REE in albid tissue, which is also the case for many other trace elements (and/or isotope studies).

Line 93:

Add citations "...support for the view that it represents a closed system for sampling past palaeo ocean chemistry (Trotter et al 2006, 2007)".

Lines 228-229:

"...substantiate the view of Trotter et al..." please replace "view" with "conclusions"
 "...which they inferred by did not demonstrate"... please replace with "based on more limited observations". Those earlier studies were not baseless or lacking evidence, as you seem to imply. "We can nevertheless.." should be "We can certainly.."

Line 244:

Add citations "conodont white matter is porous but not permeable (to within a lower limit of 150 nm)." ie. as previously concluded by Trotter et al (2006, 2007).

Line 277:

Add 2006 reference: "We evidence the view of Trotter and colleagues [6, 8]..."

Line 287:

"...chemistry over interannual timescales." I advise that "interannual" is not accurate. We are unlikely to ever (unequivocally) know the longevity of conodonts and hence the time span that their elements represent, and presently, few instruments (if any) could provide useful data at

such high-spatial resolution from those specific zones, ie. at sufficient precision within analytical uncertainties. Furthermore, we cannot rule out ontogenetic changes, which commonly occur in biominerals that also depends on the chemical element (or isotopic system) being analysed. I would therefore suggest revising this sentence to “potentially providing insights into secular changes in sea water chemistry during conodont element formation.”

Supplementary file:

Note that the supplementary table is not cited in the text. In that file, I also suggest to add one text line in the empty sheet, “throat”, stating “no throat connections discerned”.

Figures:

Can any high power images (eg. SEM) of sections through the histologies analysed be included? Such images, which are most commonly used and seen by researchers, both conodont workers and geochemists, when describing and selecting samples for analysis would be useful to compare with the images presented. This is also relevant given the complexities and confusion in recognising various types of white matter and lamellar histologies, as mentioned above.

Reviewer: 3

Comments to the Author(s)

Authors present a very interesting useful study, and potentially impactful for the general use of conodonts as geochemical bioarchives. In particular, I commend authors for obtaining remarkable data with challenging techniques. The main drawback of the manuscript is that authors draw major conclusions about conodont element permeability and crystallinity just based on partial data of two conodont elements, which are rarely used in geochemical studies. Also, some of the results are not very well explained, in particular for EBSD. I would recommend authors to address the following comments/questions:

- The general readership may not be aware of the distribution of different tissues in the two analyzed conodont elements, so it would be useful to present a general diagram with this information.
- The porosity, and subsequent permeability discussion, is only based on data from the early euconodont *Teridontus*. Do authors have any data for *Ozakodina*? For *Teridontus*, it is stated that linear throats are not visible, and the assumption is that pores are not connected. However, the section is only at the center of the element. Would a section close to the cusp or at the bottom of the element show something different? Could these linear throats present but below the technique resolution (i.e., < 100 nm)?
- Although PXCT and synchrotron tomography are not usually accessible to many researchers, EBSD is becoming a routine technique in many microscopy labs. Authors have achieved an unprecedented resolution of EBSD data for conodonts. In order to facilitate reproducibility, and wider use of this approach to the community, authors have to provide more details about the EBSD protocol under methodology (even including diagrams). Authors only state that data was achieved after 24-hour mechano-chemical polish using an OSP solution.
- The EBSD data (in Figure 2) is of very difficult interpretation. In particular, Fig. 2e is not clear what corresponds to. I would recommend showing the polished surfaces, and their correspondence to the location in the conodont element, and then the EBSD crystallographic maps matching the regions in the polishing surfaces. EBSD data without location context is not useful. Also, more interpretation of the EBSD data in terms of microstructure and element growth would be important for a better understanding of the biomineralization of conodont elements.

===PREPARING YOUR MANUSCRIPT===

===PREPARING YOUR REVISION IN SCHOLARONE===

<https://royalsociety.org/journals/authors/author-guidelines/#supplementary-material> to include a suitable title and informative caption. An example of appropriate titling and captioning may be found at https://figshare.com/articles/Table_S2_from_Is_there_a_trade-off_between_peak_performance_and_performance_breadth_across_temperatures_for_aerobic_sc_ope_in_teleost_fishes_/3843624.

Author's Response to Decision Letter for (RSOS-202013.R0)

See Appendix A.

RSOS-202013.R1 (Revision)

Review form: Reviewer 1 (Duncan Murdock)

Is the manuscript scientifically sound in its present form?

Yes

Are the interpretations and conclusions justified by the results?

Yes

Is the language acceptable?

Yes

Do you have any ethical concerns with this paper?

No

Have you any concerns about statistical analyses in this paper?

No

Recommendation?

Accept as is

Comments to the Author(s)

As I remarked in my first review of this manuscript, I believe this is an excellent piece of work from both the perspective of the technically demanding analysis and its comprehensive write up. The authors have addressed all of the minor comments I raised, as well as the possible effects of post-mortem alteration to the utility of conodonts in paleoenvironmental analyses. There has been no further discussion of the biological control of the simultaneous production of lamellar crown and syntactical crystalline white matter, but I appreciate the author's reluctance to enter into speculation, or to become hostages to fortune to the results of further experimental analysis. Furthermore, it is my opinion they have addressed the comments of the other reviewers to a similarly high standard, and I can only recommend this manuscript for publication.

Review form: Reviewer 2 (Julie Trotter)

Is the manuscript scientifically sound in its present form?

Yes

Are the interpretations and conclusions justified by the results?

Yes

Is the language acceptable?

Yes

Do you have any ethical concerns with this paper?

No

Have you any concerns about statistical analyses in this paper?

No

Recommendation?

Accept with minor revision (please list in comments)

Comments to the Author(s)

I suggest the following minor edits:

1. line 78. To be completely correct, add "...sample points and spot electron diffraction patterns." This is specifically stated and illustrated (fig 5D) in that study.
2. line 105. delete "past".
3. line 354. replace "sampling" with "analysis". Sampling is incorrect.

4. line 355. replace "Sensitive High-Resolution Ion Microprobe Probe (SHRIMP) analysis" with "techniques, such as ion microprobes.". SHRIMP isn't the only ion microprobe available, and the Cameca equivalent is (now) being used at various labs.

5. line 390. Italicise "Teridontus nakamurai"

References:

37. there are many (conventional) d18O studies prior to this. Either replace or add one or two earlier pioneering studies. (eg. Joachmiski has many)

Decision letter (RSOS-202013.R1)

Dear Dr Martinez Perez,

On behalf of the Editors, we are pleased to inform you that your Manuscript RSOS-202013.R1 "X-Ray nanotomography and Electron Backscatter Diffraction demonstrate the crystalline, heterogeneous and impermeable nature of conodont white matter" has been accepted for publication in Royal Society Open Science subject to minor revision in accordance with the referees' reports. Please find the referees' comments along with any feedback from the Editors below my signature.

Please submit your revised manuscript and required files (see below) no later than 7 days from today's (ie 24-May-2021) date. Note: the ScholarOne system will 'lock' if submission of the revision is attempted 7 or more days after the deadline. If you do not think you will be able to meet this deadline please contact the editorial office immediately.

on behalf of Professor Elizabeth Harper (Associate Editor) and Peter Haynes (Subject Editor)
openscience@royalsociety.org

Reviewer comments to Author:

Reviewer: 1

Comments to the Author(s)

As I remarked in my first review of this manuscript, I believe this is an excellent piece of work from both the perspective of the technically demanding analysis and its comprehensive write up. The authors have addressed all of the minor comments I raised, as well as the possible effects of post-mortem alteration to the utility of conodonts in paleoenvironmental analyses. There has been no further discussion of the biological control of the simultaneous production of lamellar crown and syntactical crystalline white matter, but I appreciate the author's reluctance to enter into speculation, or to become hostages to fortune to the results of further experimental analysis. Furthermore, it is my opinion they have addressed the comments of the other reviewers to a similarly high standard, and I can only recommend this manuscript for publication.

Reviewer: 2

Comments to the Author(s)

I suggest the following minor edits:

1. line 78. To be completely correct, add "...sample points and spot electron diffraction patterns." This is specifically stated and illustrated (fig 5D) in that study.
2. line 105. delete "past".
3. line 354. replace "sampling" with "analysis". Sampling is incorrect.
4. line 355. replace "Sensitive High-Resolution Ion Microprobe Probe (SHRIMP) analysis" with "techniques, such as ion microprobes.". SHRIMP isn't the only ion microprobe available, and the Cameca equivalent is (now) being used at various labs.
5. line 390. Italicise "Teridontus nakamurai"

References:

37. there are many (conventional) d18O studies prior to this. Either replace or add one or two earlier pioneering studies. (eg. Joachmiski has many)

===PREPARING YOUR MANUSCRIPT===

While not essential, it will speed up the preparation of your manuscript proof if you format your references/bibliography in Vancouver style (please see

<https://royalsociety.org/journals/authors/author-guidelines/#formatting>). You should include DOIs for as many of the references as possible.

===PREPARING YOUR REVISION IN SCHOLARONE===

<https://royalsociety.org/journals/authors/author-guidelines/#data>. You should ensure that you cite the dataset in your reference list. If you have deposited data etc in the Dryad repository,

please only include the 'For publication' link at this stage. You should remove the 'For review' link.

Author's Response to Decision Letter for (RSOS-202013.R1)

See Appendix B.

Decision letter (RSOS-202013.R2)

Dear Dr Martinez Perez,

I am pleased to inform you that your manuscript entitled "X-Ray nanotomography and Electron Backscatter Diffraction demonstrate the crystalline, heterogeneous and impermeable nature of conodont white matter" is now accepted for publication in Royal Society Open Science.

Please remember to make any datasets or code libraries 'live' prior to publication, and update any links as needed when you receive a proof to check - for instance, from a private 'for review' URL to a publicly accessible 'for publication' DOI. It is good practice to also add data sets, code and other digital materials to your reference list.

Please see the Royal Society Publishing guidance on how you may share your accepted author manuscript at <https://royalsociety.org/journals/ethics-policies/media-embargo/>. After publication, some additional ways to effectively promote your article can also be found here

<https://royalsociety.org/blog/2020/07/promoting-your-latest-paper-and-tracking-your-results/>.

on behalf of Professor Elizabeth Harper (Associate Editor) and Peter Haynes (Subject Editor)
openscience@royalsociety.org

Appendix A

Thursday March 25, 2021

Dear Editors

Re: manuscript ID RSOS-202013 "X-Ray nanotomography and Electron Backscatter Diffraction demonstrate the crystalline, heterogeneous and impermeable nature of conodont white matter"

We have revised our manuscript in light of the referees' reports. Below we provide a point-for-point response to the points that they raised, indicating where we have followed their guidance, as well as providing our reasoning in instances where we have chosen not to follow their advice.

We look forward to the outcome of your consideration of this revised manuscript.

Yours sincerely,

Carlos Martinez-Perez and colleagues

Associate Editor Comments to Author (Professor Elizabeth Harper):

Comments to the Author:

This is potentially a very interesting study. It is technically difficult and the results are significant advances. All three reviewers bring up relevant points which should all be addressed. In particular the need for:

- better description of EBSD preparation such that the findings might be reproducible

We have addressed this in our revision.

- inclusion of Ozarkodina data for all methods not just EBSD

We have used representative samples of these tissues whose homology has already been established (see e.g. Donoghue 1998 Phil Trans R Soc). Thus, there is no practical or logical need to undertake the different analyses on the same taxa. This is fortunate since it is not practically possible to undertake these same analyses on different samples; each scan requires more than a day to complete and we would be unlikely to obtain this synchrotron beamtime without an adequate scientific justification.

- appropriate recognition of previous work

We have addressed this in our revision.

It would be interesting if the findings could be, even if briefly, considered as sources of further or new information about the mechanisms of element biomineralization.

We have addressed this in our revision.

Reviewer comments to Author:

Reviewer: 1

Comments to the Author(s)

This is an excellent piece of work, both in terms of the technical challenges that have been overcome and the presentation of the results. It informs both the palaeobiology of an important group, and the utility of conodonts for paleoenvironmental analyses.

I think the authors could expand their discussion a little in two ways, but otherwise it is well written and comprehensive. Firstly, these findings have implications for the mechanisms for growth of conodont elements, which is still one of the big outstanding questions for this group. Could the authors provide any insight into the biological control of simultaneous production of lamellar crown and syntactical crystalline white matter? Particularly in tissue with such a low organic component. Secondly, it is common to see post-mortem breakage, abrasion, and/or tectonic shearing in conodont elements (even with low CAI), could the authors comment on the potential likely effects of these on the porosity and permeability, and thus inform best practice for selecting specimens for analysis?

We have addressed these points in our revision. However, there is a limit on what can be said without entering into the realms of speculation, which we would prefer to avoid, particularly when some of these questions are open to future experimental investigation.

In addition, I have the following minor corrections:

Two sentences in the abstract are so concise they are incomplete:

P1. L23. "yielding fundamental insights into the palaeotemperature and chemical composition" of what?

We have addressed this in our revision.

P1. L29. "reveal that while white matter is extremely porous and the pores are unconnected.". so...?

We have addressed this in our revision.

P6. L182-186. This long sentence is difficult to unpick.

We have addressed this in our revision.

Figure 2. Can you indicate the orientations of the cross sections in b, d and e.

We have addressed this in our revision.

Reviewer: 2

Comments to the Author(s)

The authors present a thorough, quantitative study of the crystalline nature of conodont crown tissues using the benefits of present-day high-resolution technologies. The combination of techniques used here provides indisputable proof of the large, single crystal nature of conodont albid tissue (aka white matter), its low permeability despite its high porosity, and its primary biological origin, which confirm the conclusions of earlier studies that are often ignored by researchers. These data have important implications for both understanding the growth of conodont mineralised tissues, and the suitability of albid tissue for geochemical analysis for palaeoenvironmental reconstructions.

This paper is worthy of publication in RSOS, however several corrections should be made (see below) and I would hope to see a slightly more nuanced assessment of prior work. This study unequivocally confirms the observations reported and conclusions drawn by Trotter and co-workers some 14 years ago. Convincingly, albeit arguably not totally conclusive, Trotter et al (2007) reported the single crystal and non-permeable yet highly porous nature of albid tissue (ie. the 2 key conclusions of the present study) from combined extinction contours and TEM spot diffraction patterns, as well as reconnaissance EBSD results, in a range of conodont species. Those data were combined with a thorough geochemical study (Trotter et al 2006) to support those authors' conclusions regarding the primary biological origin and suitability of albid tissue as an environmental archive (the 3rd conclusion of the present study), which should not simply be dismissed as "without material evidence". Additional citations in the text are also warranted.

We have addressed the wording in our revision. The macrocrystalline nature of white matter was inferred, rather than observed, in Trotter and colleagues (excellent) study. They mentioned EBSD analyses that they had conducted but presented no data.

Specific comments:

The introduction should canvas the complexities of both nomenclature and variability of conodont histologies, eg. cancellate ('true') albid crown that differs from apparent white matter (voids) in lamellar crown which have been previously discussed thoroughly by Donoghue 1998 and Trotter et al 2007. These issues lead to inconsistencies and confusion when describing and identifying crown tissues by various researchers, including in geochemistry studies. Note that line 225 in this paper has the first and only mention of the term "albid". It is important to clarify these terms and acknowledge differences, citing relevant studies for further reading. Despite the outcomes presented in earlier studies by Trotter and her colleagues, consistent with the findings of this recent submission, geochemistry studies sometimes contradict and even erroneously report those earlier results. These new data are therefore a very welcome contribution that will likely influence future conodont work, hopefully across all disciplines. It should also be noted that, given the widespread use of lamellar tissue in geochemistry studies (see below), better constraints on its permeability beyond those discerned in previous studies would also be helpful.

We have addressed this in our revision.

Line 24-25:

It is the high porosity of white matter/albid crown from its well-known 'cancellate' nature that has led many researchers to believe it is permeable (note the important distinction between porosity and permeability), despite contrary evidence and conclusions reported by Trotter and her colleagues.

Please revise the following excerpt: "...however, the crystallographic nature of the white matter and its inferred permeable structure, may make it unsuitable as a geochemical archive." change to "...however, the porosity and crystallographic nature of the white matter and its inferred permeability is disputed, which questions its suitability as a geochemical archive."

We have addressed this in our revision.

Line 26:

"We attempted to discriminate among these competing interpretations...".
Revise this phrase as it doesn't really make sense and the competing interpretations have not actually been stated or described (hence changes to line 25 above), so best to describe what you are actually doing: eg. "We attempted to better constrain these characteristics of white matter and address conflicting interpretations..."

We have addressed this in our revision.

Lines 53-55:

"enigmatic white matter that is the principal tissue targeted for geochemical analyses because of its assumed low permeability [6]".

We have addressed this in our revision.

I suggested that albid cancellate specifically (as opposed to lamellar 'white matter') is the preferred histology to target for geochemical studies, but this is rarely followed. Thus, the statement by these authors is completely incorrect so needs to be revised, eg. end the sentence at "...enigmatic white matter." Few conodont geochemical palaeoseawater reconstruction studies specifically target white matter for various reasons:

1. Oxygen isotope analyses comprise the most common geochemical studies of conodont apatite. These have mostly been bulk analyses using conventional wet chemistry techniques, which do not have the ability to discriminate different histological components, hence include both basal material and all crown histologies. Following the development of in-situ ion microprobe d18O conodont analyses (Trotter et al 2008), which enables specific histologies to be targeted, new studies are being undertaken yet none to date have specifically only targeted white tissue for palaeoseawater reconstructions. Both histologies are targeted in my studies, to allow ongoing comparison of their compositions (which have shown no measurable systematic difference), whereas 'hyaline' (lamellar tissue) is most commonly analysed by other researchers given its smooth surface which is (erroneously) believed to be better for ion probe analyses.
2. There are few laser ablation studies, since the initial work of Trotter et al 2006, which can be used for in situ trace element (LA-ICPMS) or some isotope analyses (LA-MC-ICPMS), and again do not specifically target white tissue mostly due to the spatial resolution and precision required. For REE studies, for example, hyaline tissue can only be used given the low to absent concentrations of REE in albid tissue, which is also the case for many other trace elements (and/or isotope studies).

We have addressed this in our revision.

Line 93:

Add citations "...support for the view that it represents a closed system for sampling past palaeo ocean chemistry (Trotter et al 2006, 2007)".

We have addressed this in our revision.

Lines 228-229:

"...substantiate the view of Trotter et al..." please replace "view" with "conclusions"

"...which they inferred by did not demonstrate"...please replace with "based on more limited observations". Those earlier studies were not

baseless or lacking evidence, as you seem to imply.
"We can nevertheless.." should be "We can certainly.."

We have addressed this in our revision.

Line 244:

Add citations "conodont white matter is porous but not permeable (to within a lower limit of 150 nm)." ie. as previously concluded by Trotter et al (2006, 2007).

We have addressed this in our revision.

Line 277:

Add 2006 reference: "We evidence the view of Trotter and colleagues [6, 8]..."

We have addressed this in our revision.

Line 287:

"...chemistry over interannual timescales." I advise that "interannual" is not accurate. We are unlikely to ever (unequivocally) know the longevity of conodonts and hence the time span that their elements represent, and presently, few instruments (if any) could provide useful data at such high-spatial resolution from those specific zones, ie. at sufficient precision within analytical uncertainties. Furthermore, we cannot rule out ontogenetic changes, which commonly occur in biominerals that also depends on the chemical element (or isotopic system) being analysed. I would therefore suggest revising this sentence to "potentially providing insights into secular changes in sea water chemistry during conodont element formation."

We have addressed this in our revision.

Supplementary file:

Note that the supplementary table is not cited in the text. In that file, I also suggest to add one text line in the empty sheet, "throat", stating "no throat connections discerned".

We have addressed this in our revision.

Figures:

Can any high power images (eg. SEM) of sections through the histologies analysed be included? Such images, which are most commonly used and seen by researchers, both conodont workers and geochemists, when describing and selecting samples for analysis would be useful to compare with the images presented. This is also relevant given the complexities and confusion in recognising various types of white matter and lamellar histologies, as mentioned above.

An example of SEM images of the analysis surface has been added, as well as a new Supplementary Fig to better explain the EBSD procedure

Reviewer: 3

Comments to the Author(s)

Authors present a very interesting useful study, and potentially impactful for the general use of conodonts as geochemical bioarchives. In particular, I commend authors for obtaining remarkable data with challenging techniques. The main drawback of the manuscript is that

authors draw major conclusions about conodont element permeability and crystallinity just based on partial data of two conodont elements, which are rarely used in geochemical studies. Also, some of the results are not very well explained, in particular for EBSD. I would recommend authors to address the following comments/questions:

The samples used in the analysis are neither rare or unusual. Indeed, they are among the best known in terms of conodont skeletal histology, used in previous studies by Müller and Nogami (1971) and Donoghue (1998). Whether they are commonly used in geochemical studies is moot; they are representative samples of the tissues that are the focus of the study, almost by definition.

- The general readership may not be aware of the distribution of different tissues in the two analyzed conodont elements, so it would be useful to present a general diagram with this information.

The two figured specimens are imaged in transmitted light explicitly to demonstrate the distribution of white matter (black) and lamellar crown tissue (translucent). Therefore, in place of adding a figure which could not be more informative, we have augmented the figure description to achieve the aim of the referee's point.

- The porosity, and subsequent permeability discussion, is only based on data from the early euconodont *Teridontus*. Do authors have any data for *Ozakodina*? For *Teridontus*, it is stated that linear throats are not visible, and the assumption is that pores are not connected. However, the section is only at the center of the element. Would a section close to the cusp or at the bottom of the element show something different? Could these linear throats present but below the technique resolution (i.e., < 100 nm)?

The tomographic data from whole specimens indicates that the white matter is homogeneous along the axis of the cusp of *Teridontus* elements.

As we describe in our revised manuscript, there is a technical limitation to the detector used in ptychographic nanotomography such that the maximum sample size cannot not exceed 100 microns in diameter, limiting our analysis to the top of the equant cusp of *Teridontus*. This, precludes us undertaking comparative analyses of *Ozakodina* – where the strongly ellipsoid cross-section of the cusp will impair good characterization of the white matter (the strong contrast in thickness would make it difficult to optimise energy). As we describe in the manuscript, at nanometric resolution we cannot formally preclude permeability, but it clearly does not occur at the scale envisaged by previous authors.

- Although PXCT and synchrotron tomography are not usually accessible to many researchers, EBSD is becoming a routine technique in many microscopy labs. Authors have achieved an unprecedented resolution of EBSD data for conodonts. In order to facilitate reproducibility, and wider use of this approach to the community, authors have to provide more details about the EBSD protocol under methodology (even including diagrams). Authors only state that data was achieved after 24-hour mechano-chemical polish using an OSP solution.

We have addressed this in our revision, including addition of a supplementary figure.

- The EBSD data (in Figure 2) is of very difficult interpretation. In particular, Fig. 2e is not clear what corresponds to. I would recommend showing the polished surfaces, and their correspondence to the location in the conodont element, and then the EBSD

crystallographic maps matching the regions in the polishing surfaces. EBSD data without location context is not useful. Also, more interpretation of the EBSD data in terms of microstructure and element growth would be important for a better understanding of the biomineralization of conodont elements.

We have included SEM images showing the polished surface preparation; unfortunately, we could just provide images for one of the specimens. In order to help the readers, we have better explain the different views and correspondences between the elements figured.

Appendix B

Thursday March 25, 2021

Dear Editors

Re: manuscript ID RSOS-202013 "X-Ray nanotomography and Electron Backscatter Diffraction demonstrate the crystalline, heterogeneous and impermeable nature of conodont white matter"

We have revised our manuscript in light of the referees' reports. All changes (minor) proposed by reviewer 2 have been addressed, including all his/her suggestions including new references.

Yours sincerely,

Carlos Martinez-Perez and colleagues